# The Role of αvβ3 Integrin in Lamina Cribrosa Cell Mechanotransduction in Glaucoma

**DOI:** 10.3390/cells13171487

**Published:** 2024-09-05

**Authors:** Mustapha Irnaten, Ellen Gaynor, Colm O’Brien

**Affiliations:** 1Clinical Research Centre, School of Medicine, University College Dublin, D04 V1W8 Dublin, Ireland; ellegaynor96@gmail.com (E.G.); cobrien@mater.ie (C.O.); 2Department of Ophthalmology, Mater Misericordiae University Hospital, D07 R2WY Dublin, Ireland

**Keywords:** integrin αVβ3, glaucoma, fibrosis, lamina cribrosa, stiffness

## Abstract

**Purpose:** Glaucoma, one of the leading causes of irreversible blindness, is a common progressive optic neuropathy characterised by visual field defects and structural changes to the optic nerve head (ONH). There is extracellular matrix (ECM) accumulation and fibrosis of the lamina cribrosa (LC) in the ONH, and consequently increased tissue stiffness of the LC connective tissue. Integrins are cell surface proteins that provide the key molecular link connecting cells to the ECM and serve as bidirectional sensors transmitting signals between cells and their environment to promote cell adhesion, proliferation, and remodelling of the ECM. Here, we investigated the expression of αVβ3 integrin in glaucoma LC cell, and its effect on stiffness-induced ECM gene transcription and cellular proliferation rate in normal (NLC) and glaucoma (GLC) LC cells, by down-regulating αVβ3 integrin expression using cilengitide (a known potent αVβ3 and αVβ5 inhibitor) and β3 integrin siRNA knockdown. **Methods:** GLC cells were compared to age-matched controls NLC to determine differential expression levels of αVβ3 integrin, ECM genes (Col1A1, α-SMA, fibronectin, vitronectin), and proliferation rates. The effects of αVβ3 integrin blockade (with cilengitide) and silencing (with a pool of four predesigned αVβ3 integrin siRNAs) on ECM gene expression and proliferation rates were evaluated using both reverse transcription quantitative polymerase chain reaction (RT-qPCR) and Western blotting in the human NLC cells cultured on soft (4 kPa) and stiff (100 kPa) substrate and in GLC cells grown on standard plastic plates. **Results:** αVβ3 integrin gene and protein expression were enhanced (*p* < 0.05) in GLC cells as compared to NLC. Both cilengitide and siRNA significantly reduced αVβ3 expression in GLC. When NLC were grown in the stiff substrate, cilengitide and siRNA also significantly reduced the increased expression in αVβ3, ECM components, and proliferation rate. **Conclusions:** Here, we provide evidence of cilengitide- and siRNA-mediated silencing of αVβ3 integrin expression, and inhibition of ECM synthesis in LC cells. Therefore, αVβ3 integrin may be a promising target for the development of novel anti-fibrotic therapies for treating the LC cupping of the ONH in glaucoma.

## 1. Introduction

Glaucoma, defined as a chronic, progressive optic neuropathy, is one of the leading causes of irreversible vision loss and blindness worldwide [1]. Glaucoma is characterized by optic nerve head (ONH) cupping, and retinal ganglion cell (RGC) death, resulting in clinical manifestations involving progressive, irreversible visual field loss. By the year 2040, it is expected that there will be approximately 118.8 million glaucoma patients globally [2].

The primary modifiable risk factor for primary open-angle glaucoma (POAG) development and progression is elevated intraocular pressure (IOP) [3]. Similarly, the only pharmacological treatment modalities available currently for POAG patients are pressure-lowering eye drops [4]. This blunt therapeutic approach is often sub-optimal and ineffective in preventing both disease progression and downstream disease sequelae including visual field defects and irreversible visual loss. This is clearly evidenced by the fact that 20% of POAG patients demonstrate disease progression despite satisfactory IOP measurements [5].

The lamina cribrosa (LC) is a structure located at the ONH that has been identified as a crucial region of injury and malfunction in glaucoma [6]. The LC is a three-dimensional, fenestrated, sieve-like connective tissue membrane that is formed by a multi-layered network of load-bearing trabeculae [7]. The LC functions to provide structural support to the unmyelinated RGC axons that pass through before exiting the eye [8]. In glaucoma, the LC changes may compress the axons passing through the laminar pores, resulting in blockade of the axonal flow [6]. Reports have clearly demonstrated an increase of ECM with age in the human optic nerve head LC [9]. As we age, the ability of the LC to constantly withstand these stresses and strains falters, resulting in biomechanical stresses [9,10,11]. The glaucomatous LC is a biomechanically stiffer structure when compared to the normal LC, due to its reduced ability to resist deformation [12]. Zeimer et al. examined post-mortem glaucomatous eyes and observed that the ONH was less compliant and mechanically stiffer in glaucomatous donor eyes when compared to the control group [9,10,13,14].

In glaucoma, the increase of ECM in the human optic nerve head LC has been investigated by Quigley, Hernadez [14,15,16,17,18,19], and others. Mounting evidence suggest that the TM from glaucomatous eyes is stiffer compared to that from age-matched normal eyes [20,21,22,23,24].

This pathological increase in tissue stiffness results in a pro-fibrotic feed-forward loop involving excessive ECM deposition, LC connective tissue fibrosis, and LC remodelling, thereby enhancing tissue stiffness even further [12]. LC stiffness and pathological glaucoma-induced LC fibrosis is governed by mechanotransduction pathways using a variety of cellular processes including the integrin family [12]. Communication between cells and the extracellular environment is often directed by several classes of cell surface receptors. Perhaps the best-characterized family of receptors involved in in-and-out directions is the integrins [25,26]. How integrins sense and respond to changes in ONH stiffness is likely to contribute to the progression of glaucoma.

Integrins are cell surface glycoproteins consisting of heterodimeric α and β subunits, and over 24 different functional heterodimeric integrin receptors have been identified. Their specific ligand binding ability enables cells to connect with their surrounding ECM, thereby controlling cytoskeleton organization that affects essential cellular functions such as cell motility, invasion, and proliferation [27]. Integrins represent a physical connection between the inside and the outside of a cell, which allows for bidirectional sensing of signals. Integrins are key elements of mechanotransduction pathways that communicate with and are regulated by the focal adhesion complex that includes FAK, c-Src, and paxillin, and proteins mediating cytoskeletal remodelling [28]. They also provide the key link between the ECM and actin cytoskeleton [29,30,31]. It has also been shown that integrins are key regulators of tissue fibrosis in numerous diseases including pulmonary, cardiac, and hepatic fibrosis [32,33,34,35]. In addition to controlling a range of physiological functions, integrins also sense ECM-induced extracellular changes during pathological events such as fibrosis, cancer, and wound healing, leading to cellular responses, which promote matrix remodelling [36].

Importantly, it has been demonstrated that integrins are overexpressed in the trabecular meshwork (TM) and ONH in glaucoma [37,38]. αVβ3 integrin regulates a number of cellular processes in the eye. In the TM, αVβ3 integrin modulates ECM assembly, actin cytoskeleton arrangement, and the formation of cross-linked actin networks (CLANS) [39,40,41]. αVβ3 integrin is involved in growth factor signalling, in particular connective tissue growth factor (CTGF) and transforming growth factor-β (TGF-β) signalling [42], including TGFβ activation and the regulation of its expression [41,43]. αVβ3 integrin also modulates the eNOS/NO pathway, a key player in outflow facility [44,45].

Importantly, αVβ3 integrin can be activated by factors including stretch, CTGF, dexamethasone, and shear stress, thereby highlighting its mechanosensitive property and potential [43,46,47]. Furthermore, the connection of integrins to the actin cytoskeleton is an inherent link necessary for the generation of contractile forces for correct fibril formation to occur [48]. Critically, increased pathological ECM deposition is observed only in cells that express a constitutively active form of αVβ3 integrin [41].

Our group has previously demonstrated that glaucomatous LC fibroblasts are activated by mechanisms such as transforming growth factor beta (TGF-β) [49], increased intracellular calcium, hypoxia, oxidative stress, and mitochondrial dysfunction [12,50]. Glaucomatous LC cells also significantly over-express several pro-fibrotic ECM genes [51], and the cells are also biomechanically stiffer in glaucoma [52].

Furthermore, both normal and glaucomatous LC fibroblasts demonstrate mechanosensitive properties, and we have shown that an increase in matrix stiffness singularly evokes a myofibroblastic, pro-fibrotic, glaucomatous phenotype in normal LC cells [52]. We hypothesise that increased tissue stiffness drives LC fibroblast to myofibroblast differentiation leading to ECM remodelling, activating numerous signal transduction pathways including integrin-mediated mechanotransduction. In the current study, we investigated the functional and molecular roles of αVβ3 integrin in glaucoma LC cells and examined whether its expression level parallels ECM gene expression levels and cellular proliferation rates in normal LC cells grown on stiff matrix and in glaucoma LC cells grown on standard flat-bottom plastic plates.

## 2. Materials and Methods

### 2.1. Cell Culture and LC Cells Characterization

In total, five eyes from donors with no prior history of glaucoma and five eyes from donors diagnosed with glaucoma (Table 1) were obtained from the Central Florida (Tampa) Eye Bank for Transplant and Research within 24 h post-mortem. The normal and glaucoma groups are age-matched, with ranges of 79 to 88 years (non-glaucoma) and 79 to 86 years (glaucoma), and have a comparable average age, 85.8 years and 83.6 years, respectively. Glaucomatous eyes were confirmed based on retrieved medical records and/or confirmation of previous glaucoma diagnosis by family members. All eyes were obtained and managed in compliance with the Declaration of Helsinki for research involving human tissue. All eyes were cut flush with the posterior sclera and were imaged from the posterior cut surface. The optic nerves of all specimens were examined, and the specimens were further classified.

The isolation of LC cells from a single ONH explant from human donor eyes has been described in great detail by NN Lopez et al., 2020 [53]. The LC cells were isolated from the LC removed from donor optic nerves. It is certainly possible that some of the LC cells have migrated from the peri-papillary sclera connective tissue. The isolated LC cells showed morphological features expected of human LC cells [54].

LC cells were cultured at 37 °C in Dulbecco’s modified Eagle’s medium (DMEM) (Sigma, Dublin, Ireland) supplemented with 10% heat inactivated foetal bovine serum (FBS) and 1% L-glutamine and 2% penicillin–streptomycin (Sigma, Dublin, Ireland) under humidified air (5% CO_2_). Characterisation of these cells was performed as described previously [54,55]. Cells expressing a fibroblast marker, α-smooth muscle actin (α-SMA) that did not express an astrocyte marker, glial fibrillary acidic protein (GFAP), and a microglial marker, ionized Ca^2+^ binding adapter molecule 1 (Iba1), were characterised as LC cells [54,55]. Cells were maintained in full medium, which was replaced every 48 h. Cells were plated at 70–80% confluence and passaged every 3 days following 0.25% trypsin/EDTA (Sigma, Dublin, Ireland) digestion, and used until passage 8.

### 2.2. LC Cells of Normal Donors Cultured on Stiffened Matrices

LC cells were seeded at a density of 10^4^ cells/cm^2^ on commercially available Collagen-1 coated polyacrylamide hydrogel matrix (Softwell Collagen, Matrigen Products, Cell Guidance Systems, Cambridge, UK) of matrix stiffness 4 kPa (normal physiological LC stiffness) and 100 kPa (representing glaucomatous LC stiffening), respectively, for 3 to 4 days. Once 80% confluent, RNA and proteins were extracted from LC cells either treated with cilengitide, β3 integrin siRNA, scrambled siRNA, or untreated controls using the RNA isolation protein extraction protocols described below.

### 2.3. Cilengitide Treatment

Cilengitide was obtained from Merck (Dublin, Ireland). Cilengitide was dissolved in dimethyl sulfoxide (DSMO) to the required concentration of 100 μM. Cells (1 × 10^6^ cells) were plated in 6-well plates on 4 kPa (soft) matrix and 100 kPa (stiff) matrix. Once 80% confluent, cells were serum-starved for 24 h and were either treated in triplicates with cilengitide (10 µM) (treated group) or not treated (control group). Out of each 6-well plate (4 kPa and 100 kPa), 3 wells received cilengitide treatment, and 3 wells remained untreated controls. Five normal control donors and 5 glaucoma donors were used in this study. Negative controls consisted of untreated samples containing only the vehicle (0.01% DMSO). RNA and/or protein were isolated from both untreated control cells and cilengitide-treated cells using the RNA and protein isolation protocols described below.

### 2.4. β3 Integrin siRNA Transfection

A mixture of four pooled β3 integrin small interfering RNAs (siRNAs) (siRNA-SMART pool) and the negative control (siCONTROL non-targeting siRNA) were obtained from Dharmacon (Research Inc., London, UK). LC cells were seeded at equal densities of 1 × 10^6^ cells/mL of 6-well plates (3 wells per condition) and transfected at 50–60% confluence with transfection reagent (5 μL) (Dharmacon, London, UK), according to the manufacturer’s instructions. The sequences of β3 integrin siRNAs (siRNA-SMART pool) used are as follow: Sequence 1: GAAAGUCCAUCCUGUAUGU; Sequence2: GAAAAUCCGUUCUAAAGUA; Sequence 3: UUACUGCCGUGACGAGAUU; Sequence 4: CGUCUACCUUCACCAAUAU. Following 48 h of culture, cells were starved without antibiotics and FBS in 1.0 mL of Opti-MEM^®^ for 24 h prior to transfection. The human LC fibroblast cells were transfected with either 50 nM siRNA anti-human β3 integrin (siRNA-SMARTpool), or 50 nM negative control siRNA. All the sequences used for siRNAs were blasted (http://blast.ncbi.nlm.nih.gov/Blast.cgi) (accessed on 5 May 2022) to avoid silencing unrelated genes. Cells were transfected in Opti-MEM medium (Invitrogen, Paisley, UK) without penicillin or streptomycin, which would otherwise cause cell death during transfection, using a mixture of four β3 integrin siRNA reagents in “difficult-to-transfect cell lines” Lonza™ and Nucleofector^TM^ solution to overcome the limitations associated with lipid-reagent-mediated transfection. After 6 h of culture, the transfection mixture was then removed and replaced with normal growth medium and cells were grown for an additional 68 hpost-transfection according to the manufacturer’s instructions (Dharmacon Research Inc., London, UK). Cells were assayed after 72 h post-transfection, total RNA and cellular protein were extracted, and their levels were determined by real-time RT-PCR and Western immunoblotting analysis, respectively, to ensure the transfection efficiency. All transfections were repeated three times and were from 5 normal and 5 glaucoma donors.

### 2.5. Total RNA Extraction and Quantitative RT-PCR

Total RNA from all primary cultures of LC fibroblasts was extracted using Tri-Reagent© protocol (Invitrogen, Dublin, Ireland) followed by DNase I (Qiagen, London, UK) treatment. Total RNA was extracted from the lysate using standard procedures. The extracted RNA was used to synthesize the cDNA using oligo-dT primers, deoxy-nucleotides (dNTPs), and enhanced avian reverse transcriptase (eAMV) (Merck, Dublin, Ireland) cDNA, according to the manufacturer’s instructions. The samples were incubated in the Thermocycler for 60 min at 42 °C, and reverse transcriptase was heat-inactivated for 5 min at 95 °C and then immediately cooled to 4 °C and stored at −80 °C. The concentration and the purity of the RNA were determined using aNanoDrop™ 1000 Spectrophotometer (Thermo Scientific, Dublin, Ireland).

The quantitative RT-PCR was then used to determine the relative expression of the αV, β3, and β5 integrins, and fibrotic ECM genes including α-SMA, Col1A1, fibronectin, and vitronectin. For RT-PCR experiments, the results shown are from RNAs using integrin-αV and -β3 primers separately. All PCR reactions were carried out on a LightCycler^®^480 System (ROCHE Diagnostics, Dublin, Ireland), using Platinum^®^ SYBR^®^ Green qPCR Super-Mix (ROCHE, Dublin, Ireland) along with human-gene-specific forward and reverse primers (Table 1). All pair primer sequences were blasted using the primer-blast software on the NCBI website (https://www.ncbi.nlm.nih.gov/) (accessed on 5 May 2022) prior to the experiment. A master mix was made comprising SYBR^®^ Green, forward and reverses primers, and nuclease free water. Two μL of cDNA was added, and the total reaction volume was 10 μL per well of a 96-well plate in triplicates. The PCR thermal profile consisted of an initial 10 min at 95 °C, followed by 45 cycles of 95 °C for 15 s and 60 °C for 1 min. Target gene mRNA expression levels were normalized to that of housekeeping 18S gene as an internal control for quantification, and relative change was calculated with the comparative 2∆Ct method [56]. Primer sequences were obtained from Merck/Sigma Ltd., Dublin, Ireland, and their specific information is provided in Table 2. Each treatment was performed in triplicate, and the results were presented as the mean ± standard deviation from 5 normal and 5 glaucoma donors.

### 2.6. Cell Lysate Preparation and Western Blot Analysis

LC cells (10^6^ cells) were cultured in 6-well plates and serum-starved for 24 h and either treated with cilengitide (10 µM) or -β3-siRNA (50 nM) in triplicates (treated group) or not (untreated control group) for 24 h. To isolate the cellular protein, LC cells were washed with cold phosphate-buffered saline (PBS) and lysed in a 500 μL lysis radio-immune precipitation assay (RIPA) buffer (Sigma, Dublin, Ireland) containing a protease inhibitor cocktail (Sigma, Dublin, Ireland) for 30 min, and lysates were collected with a cell scraper. Protein quantification was performed using the bicinchoninic acid (BCA) protein assay (Pierce, Thermo Scientific, Dublin, Ireland). A total of 20 μg of protein were loaded onto a 10% polyacrylamide-SDS-PAGE and blotted onto nitrocellulose membranes (Merck–Millipore, Dublin, Ireland) at 0.5 V/cm^2^ for 45 min. For immuno-detection, the membranes were blocked for 1 h at room temperature with 5% dry milk in Tris-buffered saline containing 0.1% Tween-20 (TBS-T) solution and then probed overnight at 4 °C with an anti-integrin-β3 primary antibody (ab227702;1:500; Abcam, Cambridge, UK),anti-Col1A1 primary antibody (1:300; Abcam, Cambridge, UK), and anti-αSMA primary antibody (1:300; Abcam, Cambridge, UK). Membranes were then washed three times with TBS-T for 10 min and incubated with anti-mouse HRP-conjugated secondary antibody (1:10,000, Abcam, Cambridge, UK) in TBST for 1 h at room temperature. Following three rounds of washing with TBS-T for 5 min, a chemiluminescence signal was generated using ECLPlus detection kit (Fisher Scientific, Dublin, Ireland). The bands were quantified with an ImageQuant densitometric scanner (Molecular Dynamics, Sunnyvale, CA, USA). To ensure equal protein loading per lysate sample, the membranes were stripped and re-probed for β-actin (Abcam, Cambridge, UK). Each treatment was performed in triplicate (technical replicates) and from 5 different eye donors (biological replicates). The results were presented as the mean ± standard deviation.

### 2.7. Cell Viability Assay (MTT)

To assess cell viability and the possible cytotoxicity of cilengitide on LC cells, we utilised a MTT tetrazolium reduction assay (3-(4,5-dimethylthiazol-2-yl)-2,5-diphenyl tetrazolium bromide) (CT02, Millipore, Burlington, MA, USA). In preparation, 10 mL of PBS pH 7.4 was added to MTT and stored overnight in the dark at 4 °C to allow for crystals to fully dissolve. Treated/non-treated normal and glaucoma LC cells (1 × 10^4^ cells/well) (n = 5 donors per group) were seeded in triplicate into 96-well plates. After 48 h incubation, serum-complete medium was removed and cells were incubated in serum-free medium. Cells were then rinsed with PBS and were assigned into untreated (control) or treated wells. Treatment wells received 10 (μM) cilengitide for 24 h in 100 μL 10% FBS serum-rich medium, with control wells receiving 100 μL 10% FBS serum-rich medium only. Following 24 h, 10 μL of MTT was added to each well, and cells were incubated for an additional 4 h. Furthermore, negative control wells were utilised containing 100 μL of serum-complete medium (cell-free) only. Following this, 100 μL of isopropanol with 0.04 N HCl was added to each well. Absorbance was then read using a SpectraMax^®^ Plus 384 microplate (Molecular Devices, San Jose, CA, USA) spectrophotometer and accompanying software SoftMax^®^ Pro 6.2.2 (Molecular Devices, San Jose, CA, USA) at a test wavelength of 570 nm and a reference wavelength of 630 nm. As per the manufacturer’s instructions, a single absorbance reading following 4 h of MTT incubation was captured. Cell viability was presented as a percentage of each concentration relative to control. Each experiment was performed on cells from 5 donors per group, and repeated three times to ensure reproducibility of results.

### 2.8. Cell Proliferation Assay

Proliferation was assessed using the colorimetric MTS assay after 96 h of cell culture. LC cell proliferation was assessed in age-matched normal LC cells grown on soft and stiff matrix, and also in normal and glaucoma LC cells cultured on standard Costar plastic 96-well plates, in the presence (treated group) or absence (untreated control group) of cilengitide or siRNA-β3 integrin. For this, LC cells were initially seeded at a density of 10^4^ cells/well on “CellTiter 96^®^” 96-well flat-bottom plates (Promega, Dublin, Ireland), in a final volume of 100 μL/well. Cell were allowed to grow for 48 h, serum-starved for another 24 h, then either treated (in triplicate) with cilengitide (10 µM) for an additional 24 h or transfected with β3 integrin siRNA (50 nM). The level of LC cell proliferation was measured using the methyl thiazolyl tetrazolium salt (MTS) colorimetric cell counting assay according to the manufacturer’s protocol (Promega, Ireland). Twenty μL of the CellTiter 96^®^ Aqueous One Solution Reagent (Promega, Ireland) was added to each well, and the plates were incubated in a humidified 5% CO_2_ atmosphere at 37 °C for 1 h. The supernatant was removed and 150 μL/well of DMSO was added to the plates to solubilize the formazan salt crystals. Plates were then incubated for 10 min at room temperature. The plate was read at a wavelength of 490 nm on the SpectraMax spectrophotometer (Molecular Devices Corp., Wokingham, UK) and analysed using SoftMax Pro 7.1.1 Software (Molecular Devices Corp., UK). All experiments were performed in triplicates (technical replicates) and from 5 different donors per group (biological replicates), and the results were presented as the mean ± standard deviation (SD).

### 2.9. Statistical Analyses

Results were obtained for each cell culture experiment with five independent biological eye donors, and each experiment was repeated three times. Experiments were performed in LC cells derived from age-matched normal controls and glaucoma eye donors. The results were expressed as mean ± SD. Student’s *t*-tests were performed for comparisons between two groups. One-way analysis of variance (ANOVA) with Tukey’s was used to compare more than two groups. A *p* value of <0.05 was considered as statistical significance. OriginPro 7.0 software, was applied for statistical analyses of the results generated in this study (Origin Lab, Bucks, UK).

## 3. Results

### 3.1. Expression of αVβ3 and αVβ5 Integrins in LC Cells and in Human ONH

The expression levels of β3 integrin were initially analysed in normal and glaucoma LC cells using quantitative real-time RT-PCR, and the results indicated that β3 integrin was significantly relatively over-expressed in GLC cells (0.861 ± 0.055) when compared to NLC cells (0.482 ± 0.038) (*p* < 0.05, n = 5) (Figure 1A). This was confirmed by Western blot analysis, demonstrating significant increase in protein expression levels in GLC donors (6.477 ± 0.622) compared to NLC patients (3.364 ± 0.534) (*p* < 0.05, n = 5) (Figure 1B). No significant differential expression of *β5* integrin was found when comparing normal control to glaucoma LC cells (Figure 1C). Also, no significant differential expression was found for αV integrin between normal and glaucoma LC cells.

### 3.2. Dose-Dependent Effect of Cilengitide on LC Cell Viability

The dose–response effect of cilengitide was investigated on a primary culture of glaucoma LC cells. Following 24 h of serum starvation, cells were incubated with cilengitide at increasing concentrations of 0–100 μM (Figure 2). Twenty-four hours later, the cells were subjected to the viability assay. All concentrations from 0.1 µM to 10 µM exhibited viability above 96%, indicating that at these concentrations, cilengitide had no cytotoxic effects on cell viability (Figure 2). However, cell viability was decreased, reaching statistical significance at 100 μM (Figure 2). Thus, a concentration of 10 µM was used for all subsequent experiments.

### 3.3. Cilengitide Treatment Down-Regulated the Expression of β3 Integrin in Glaucoma LC Cells

We determined the effect of cilengitide on β3 integrin protein expression in normal and glaucoma LC cells by Western blot analysis. As shown in Figure 3A, the level of β3 integrin protein expression level was significantly lower in untreated normal LC cells than that of untreated glaucoma LC cells, and this enhancement was significantly reduced by cilengitide treatment of glaucoma LC cells (9.201 ± 1.246 untreated vs. 3.424 ± 0.976 treated) (*p* < 0.05, n = 5) (Figure 3A).

### 3.4. Cilengitide Treatment Down-Regulated the Stiffness-Induced Expression of β3 Integrin in Normal LC Cells

NLC cells were grown on soft (4 kPa) and stiff (100 kPa) substrates, and at 80% confluence they were either treated (treated group) or not (untreated group) with a potent inhibitor of αVβ3 integrin cilengitide (10 μM) for 24 h. Quantitative real-time RT-PCR results show that cilengitide (10 μM) significantly reduced the stiffness-induced β3 integrin gene expression (1.654 ± 0.174 on untreated stiff matrix vs. 1.071 ± 0.135 on untreated soft matrix and 1.204 ± 0.097 on treated stiff matrix) (*p* < 0.05, n = 5) (Figure 3B). To validate these data, we examined the effect of cilengitide on β3 integrin at the protein level; we found that matrix stiffness (100 kPa) induced a significant increase in β3 integrin protein expression level when compared to cells grown on soft matrix (4 kPa) (*p* < 0.05, n = 5) (Figure 3C). In addition, Western blot analysis showed that cilengitide treatment reduced β3 integrin protein expression levels on both soft (3.638 ± 0.563 untreated vs. 2.739 ± 0.247 treated) and stiff (100 kPa) matrix (8.821 ± 1.246 untreated vs. 6.081 ± 0.984 treated) (*p* < 0.05, n = 5) (Figure 3C).

### 3.5. SiRNA Knockdown Treatment Down-Regulated the Expression of β3 Integrin in Glaucoma LC Cells

Cilengitide is a known antagonist selective not only for β3 integrin but also for β5 integrin [57]. To overcome the limitations associated with cilengitide specificity, we utilised siRNA to specifically knock down β3 integrin. The successful transfection of β3 integrin siRNA or control scrambled siRNA in LC cells was confirmed via qRT-PCR 72 h post-transfection. 

As shown in Figure 4A, we examined the effect of siRNA targeting β3 integrin protein expression by Western blot analysis at day 3 post-transfection on normal and glaucoma LC cells. Integrin β3 siRNA transfection significantly reduced the β3 protein expression levels by ~74% compared to untreated LC cells (*p* < 0.05, n = 5) (Figure 4A).

### 3.6. siRNA Knockdown Treatment Inhibited the Stiffness-Induced Expression of β3 Integrin in Normal LC Cells

As shown in Figure 4, β3 integrin siRNA transfection significantly supressed the stiffness-induced β3 integrin gene transcription by ~65% when compared to untreated control LC cells (*p* < 0.05, n = 5) (Figure 4B). We also examined the effect of β3 integrin siRNA transfection on β3 integrin protein expression using Western blot analysis. The results clearly indicated that siRNA treatment had significantly down-regulated the stiffness-induced protein expression level of β3 integrin by approximately 45% compared to that of the control non-transfected group (*p* < 0.05, n = 5) (Figure 4C). Therefore, these results demonstrate that the enhanced stiffness-induced αVβ3 integrin expression was specifically suppressed by β3 integrin siRNA silencing (Figure 4B,C).

### 3.7. Cilengitide and siRNA Treatments Reduced the Expression of ECM Production in Glaucoma LC Cells

We then checked the levels of ECM gene expression following 24 h of cilengitide treatment on control and glaucoma LC cells grown on standard plastic plates. As shown in Figure 5, the ECM gene expression levels were significantly lower in untreated normal LC cells than those of untreated glaucoma LC cells for all tested genes including Col1A1, αSMA, fibronectin, and vitronectin. Cilengitide treatment of glaucoma LC cells significantly reduced the ECM gene expression of Col1A1 (from 1.022 ± 0.073 to 0.453 ± 0.064), αSMA (from 0.948 ± 0.068 to 0.239 ± 0.038), fibronectin (from 0.892 ± 0.067 to 0.198 ± 0.039), and vitronectin (from 0.877 ± 0.065 to 0.283 ± 0.071) (*p* < 0.05, n = 5) (Figure 5). Glaucoma LC cells were also transfected with siRNA targeting β3 integrin and the results, at day 3 post-transfection, showed a clear significant inhibition of ECM gene transcription for Col1A1 (from 1.022 ± 0.073 to 0.168 ± 0.035), αSMA (from 0.948 ± 0.068 to 0.097 ± 0.022), fibronectin (from 0.892 ± 0.067 to 0.134 ± 0.0328), and vitronectin (from 0.877 ± 0.065 to 0.075 ± 0.021) (*p* < 0.05, n = 5) (Figure 5).

### 3.8. Cilengitide and siRNA Treatments Reduced the Stiffness-Induced Expression of ECM in Normal LC Cells Grown on Stiff (100 kPa) Substrate

ECM gene transcription has been shown to depend on integrin activation [58]. In line with this, we found that normal LC cells grown on stiff matrix and glaucoma LC cells grown on standard plastic plates both express enhanced β3 and ECM production. We first measured the expression levels of ECM genes in NLC cells grown on soft and stiff matrix following 24 h treatment with cilengitide (10 µM). Cilengitide reduced the stiffness-induced expression of Col1A1 (from 0.872 ± 0.097 stiff untreated to 0.364 ± 0.039 stiff treated), αSMA (from 0.982 ± 0.115 stiff untreated to 0.356 ± 0.044 stiff treated), fibronectin (from 0.763 ± 0.105 stiff untreated to 0.212 ± 0.036 stiff treated), and vitronectin (from 0.791 ± 0.108 stiff untreated to 0.175 ± 0.053 stiff treated) (*p* < 0.05, n = 5) (Figure 6). Moreover, as cilengitide is an antagonist that selectively inhibits not only β3 integrin but also β5 integrin [59], we further examined the specific contribution of β3 integrin in pro-fibrotic ECM gene expression in normal LC cells grown on stiff matrix by silencing β3 integrin using siRNA. Thus, we tested the effect of siRNA β3 integrin silencing on ECM gene expression at day 3 post-transfection in NLC cells grown on stiff matrix. As shown in Figure 6, stiffness induced a significant (*p* < 0.05) enhancement, while silencing β3 integrin significantly suppressed the stiffness-induced enhancement in gene transcription of Col1A1 (from 0.872 ± 0.097 stiff untreated to 0.174 ± 0.019 stiff treated), αSMA (from 0.982 ± 0.115 stiff untreated to 0.117 ± 0.015 stiff treated), fibronectin (from 0.763 ± 0.105 stiff untreated to 0.191 ± 0.017 stiff treated), and vitronectin (from 0.791 ± 0.108 stiff untreated to 0.183 ± 0.013 stiff treated) (*p* < 0.05, n = 5) (Figure 6).

### 3.9. Cilengitide and siRNA Knockdown of αVβ3 Integrin Inhibited Proliferation in Glaucoma LC Cells

To follow up, we compared the proliferation rate between control and glaucoma LC cells grown on standard plates (approx. 10^6^ kPa) in the presence or absence of cilengitide or siRNA anti-β3 integrin. For this, LC cells were seeded at a density of 10^4^ cells/well in a 96-well plate. After the cells were cultured for 48 h, they were serum-starved for another 24 h before treatment with cilengitide (10 µM) for an additional 24 h. Results showed a significant increase in cell proliferation rate in untreated glaucoma LC cells (187.31 ± 21.14% vs. 135.46 ± 14.70 in untreated normal LC cells), while cilengitide treatment of glaucoma cells significantly reduced the cell proliferation rate to 116.182 ± 12.42% (*p* < 0.05, n = 5) (Figure 7A). In addition, siRNA targeting β3 integrin also significantly reduced LC cell proliferation rate in glaucoma from 187.31 ± 21.14% in untreated glaucoma compared to 83.42 ± 8.44% in treated glaucoma LC cells (*p* < 0.05, n = 5) (Figure 7A).

### 3.10. Cilengitide and siRNA Anti-β3 Integrin Inhibited the Stiffness-Induced Proliferation in Normal LC Cells

The enhanced β3 integrin expression levels in stiffer matrix and its impact on ECM gene transcription regulation encouraged us to test whether β3 integrin contributes to the stiffness-induced cellular proliferation in LC fibroblast cells. There is also a growing body of evidence reporting that β3 integrin silencing plays a key role in mediating proliferation in fibrosis diseases including cardiac and pulmonary fibrosis [27]. Therefore, we examined the possible contribution of β3 integrin on LC cell proliferation following β3-targeting siRNA transfection. MTS assay was assessed in normal LC cells grown on soft (4 kPa) or stiff (100 kPa) matrix, either in the presence (treated group) or absence (untreated group) of cilengitide (10 µM) or β3 integrin siRNAs (50 nM). As shown in Figure 7B, the proliferation rate was significantly elevated in normal LC cells grown on stiff (100 kPa) matrix (untreated stiff 171.61 ± 16.59% vs. untreated soft 122.82 ± 10.73%), and cilengitide treatment significantly reduced cell proliferation rate (stiff untreated 171.61± 16.59% vs. stiff treated 97.32 ± 8.70%) (*p* < 0.05, n = 5) (Figure 7B).

siRNA integrin β3 transfection also significantly inhibited the stiffness-induced proliferation rate (stiff untreated 171.61 ± 16.59% vs. stiff treated 72.63 ± 7.94%) (*p* < 0.05, n = 5) (Figure 7B).

## 4. Discussion

It is essential that we identify the underlying cellular and biological mechanisms that drive fibrosis in glaucoma in order to develop targeted therapies aimed at inhibiting these fibrotic mechanisms, thereby preventing or halting the development of the clinical manifestations of glaucoma. We hypothesize that increased tissue stiffness drives LC fibroblast to myofibroblast differentiation and ECM fibrosis, activating numerous signal transduction pathways including mechanotransduction. 

In this study, we sought to elucidate the role of αVβ3 integrin in glaucoma. We found over-expression of the αVβ3 integrin in glaucoma and in normal LC fibroblast cells grown on stiffer matrix (100 kPa) compared to cells from normal donors. The over-expression was suppressed by cilengitide and αVβ3 integrin siRNA treatments. Blockage or silencing of αVβ3 integrin also resulted in a significant decrease in ECM production and cellular proliferation rates. Collectively, these results show that the αVβ3 integrin has a beneficial effect on ECM and LC cellular proliferation level. Therefore, cilengitide treatment or siRNA-mediated knockdown of the αVβ3 integrin could be considered as a novel therapeutic tool for cupping of the optic nerve in glaucoma treatment. 

In recent years, several bodies of work have aimed to uncover the therapeutic benefits of αvβ3 integrin inhibition in several ophthalmic diseases, including glaucoma. A number of studies carried out in trabecular meshwork by Filla et al. [60] suggest that αVβ3 integrin activation engages a downstream pro-fibrotic pathogenic mechanism that drives POAG pathogenesis. It is well established that the pro-fibrotic cytokine TGF-β is over-expressed in the aqueous humour, trabecular meshwork [60], and the ONH in glaucoma [58,61]. TGF-β activation may be mediated by integrins, and that integrin-mediated TGF-β over-activation may be an inherent driving factor in the pathogenesis of glaucoma [49,51,62,63]. αV integrins have been demonstrated as propagators of TGF-β activation in normal and cancerous tissues [64]. αVβ3 integrin has been shown to drive fibroblast contraction and strain stiffening of soft provisional matrix in fibrosis [35]. Data obtained from RNA-sequencing analyses of the TM of POAG patients suggest that over-expression of the pro-fibrotic growth factor TGFβ2 in the glaucomatous TM may be stimulated and propagated by αVβ3 integrin activation [37]. Furthermore, it is postulated that αVβ3 integrin over-expression mediates pathogenic fibronectin fibrillo-genesis, thereby resulting in consequent activation of downstream integrin-mediated signalling pathways [37,60].

Faralli et al. demonstrated that dexamethasone significantly increases and prolongs the expression of αVβ3 integrin in TM cells through a secondary glucocorticoid response mechanism involving the calcineurin/NFAT signalling pathway [46]. This group has also correlated the expression of the αVβ3 integrin and IOP in mice models. They found that αVβ3 integrin expression was associated with raised IOP, and that β3 subunit inhibition resulted in lower IOP two weeks after treatment [65]. Morrison’s group compared normal, rhesus monkey, and glaucomatous ONH and found that α2, α3, β1, and β4 integrins were significantly more abundant in the region anterior to the LC of glaucomatous ONH when compared to normal donors [38]. This important body of work highlighted a potential pro-pathogenic role of integrins in glaucoma ONH fibrosis pathogenesis, and supports the concept that integrins sense and respond to stress and strain in the LC, thereby making them possible key players intrinsically involved in the ONH biomechanical alterations that drive POAG pathogenesis [38].

Furthermore, it has been evidenced that αVβ3 integrin is involved in other neurodegenerative diseases other than glaucoma. It has been demonstrated that αVβ3 integrin was up-regulated in retinal ganglion cells and the glial cells of the optic nerve head after nerve crush in mice [23]. In addition, the expression levels of αvβ3 are also elevated in reactive astrocytes [66]. αvβ3 is a receptor for thymus cell antigen 1 (Thy1) [67]. Thy1, also called Cluster of Differentiation 90 (CD90), is a glycophosphatidylinositol-anchored glycoprotein that is implicated in neuronal injury [68,69]. αvβ3 is shown to be a modulator of the signalling cascade to influence astrocyte-to-neuron communication and astrocyte adhesion and migration [70,71,72,73].

In addition, integrins are also well characterized in the cornea. Corneal epithelial and endothelial cells have been shown to express integrins on their surface. In the central cornea, α_2_β_1_, α_3_β_1_, α_V_β_5_, and α_6_β_4_ integrins are located within the epithelium, with the highest expression level in the basal cells. The α_6_β_4_ integrin mediates adhesion to the epithelial basement membrane (EpBM) using hemidesmosomes, while α_3_β_1_ and α_V_β_5_ involve focal adhesions which are actin-based. The integrins expressed at the EpBM can mediate adhesion of corneal epithelial cells to fibronectins, vitronectin, collagens, and laminins. Cells are less proliferative and adhesive to the basement membrane when integrin expression decreases [74].

Integrins play a key role in normal eye development and also in the development of pathological processes, such as injuries of the cornea, keratoconus, allergic eye disease, keratitis, and dry eye disease [75]. For example, the αvβ6 integrin is a key player in corneal fibrosis [76]; integrins α1, α3, α4, αL, β1, β3, and β4 were up-regulated in the hereditary eye disease Fuchs’ corneal dystrophy [77]. The αL integrin plays a vital role in dry eye diseases, and its inhibition significantly improves ailments [78]. Whether the corneal integrin expression changes in glaucoma is currently unknown but worthy of investigation.

For the role of cilengitide as a therapeutic target in glaucoma, to the best of our knowledge, there is no mention of cilengitide in published articles looking at the role of systemic medications in glaucoma. For example, in a review article by Wu A. et al. (2020) [76], it was demonstrated that various commonly used systemic medications can affect optic nerve perfusion, retinal ganglion cell survival, and aqueous humour outflow facility, suggesting that these medications may modulate risk for OAG or disease progression. Over the past decade, a number of these drugs have been found in large retrospective population-based studies to potentially modulate the risk of POAG, including metformin, statins, beta blockers, calcium channel blockers, SSRIs, bupropion, and postmenopausal hormones, although discordant findings exist in the literature, highlighting the need for well-designed prospective randomized controlled trials to best examine the relationships between these systemic medications and risk for glaucoma. Within this updated list of systemic medications that may modulate the risk of glaucoma, cilengitide was not mentioned.

The role of cilengitide as a therapeutic target for other pathological forms of fibrosis of organ systems, other than in the eye, has been studied, yielding promising results. Using a mouse model to study hypertensive-driven cardiac fibrosis, Perrucci et al. demonstrated that cilengitide treatment decreased the expression of pro-fibrotic ECM genes Col1A1 and αSMA [79]. Furthermore, cilengitide decreased the differentiation of fibroblasts to myofibroblasts, thus attenuating fibrosis. This study supports the concept that cilengitide may behave as a novel anti-fibrotic therapy [79]. Similarly, results from Bouvet et al. firstly identify αV integrin as an important regulator of cardiac fibrosis, and importantly, pharmacological inhibition of αV using cilengitide resulted in reduced cardiac fibrosis, with an overall improvement in cardiac function [80]. Other reports indicated that β3 integrin plays a critical role in cardiac hypertension fibroblasts [81]. Cilengitide has also been proposed as a treatment for systemic sclerosis (SSc). Bagnato et al. analysed murine models of SSc and found that cilengitide treatment reduced pro-fibrotic gene expression of α-SMA and TGF-β1 and attenuated renal glomeruli collapse. Thus, to assess the effect of anti-fibrotic treatment in glaucoma, we used cilengitide, a potent inhibitor of αVβ3 integrin as recognized in human cancer studies [82]. Cilengitide was expected to be beneficial both due to its specific anti-angiogenic effect in vitro and in vivo [34,83], and to its anti-proliferative and anti-fibrotic activity in vitro [84].

In this study, we found that β5 integrin is expressed at very low levels without differential expression when comparing normal LC fibroblast cells with glaucoma LC fibroblast cells. Thus, we investigated whether specifically reducing the levels of αVβ3 integrin gene and protein expression in the established human LC cell line, in which this gene and protein were overexpressed, might result in a decrease in the excessive production of ECM and LC cell proliferation levels. Our results demonstrated that the stiffness-induced enhancement in ECM production and cell proliferation rate were suppressed when silencing LC cells with an anti-β3 integrin siRNA. These results indicate a strong relationship between expression of the αVβ3 integrin and ECM production and proliferation rate in glaucoma and in normal LC fibroblast cells grown on stiffer matrix. According to the literature, the αVβ3 integrin gene is strongly associated with the equilibrium of ECM production and cell proliferation rate, and it functions in the same manner in glaucoma LC cells.

Herein, we report for the first time that siRNA can selectively and efficiently silence the expression of the integrin β3 subunit in glaucoma LC cells. We showed that integrin β3 silencing affects the ECM production and proliferation rate of glaucoma LC cells.

## 5. Conclusions

In conclusion, our results indicate that cilengitide and β3 integrin siRNA function to disrupt the pathogenic integrin-mediated stiffness cycle that causes excessive pro-fibrotic ECM deposition in the ONH in glaucoma. Therefore, αVβ3 integrin could be regarded as a gene that promotes fibrosis in glaucoma, and regulation of its expression via cilengitide or siRNA silencing would be an interesting and novel therapeutic target for the treatment of glaucoma. 

## 6. Limitations of the Study

The limitations of using human primary cultured cells include the lack of availability of cell donors and authors do not have information on what treatment the donors had or the severity of the glaucoma.

## Figures and Tables

**Figure 1 cells-13-01487-f001:**
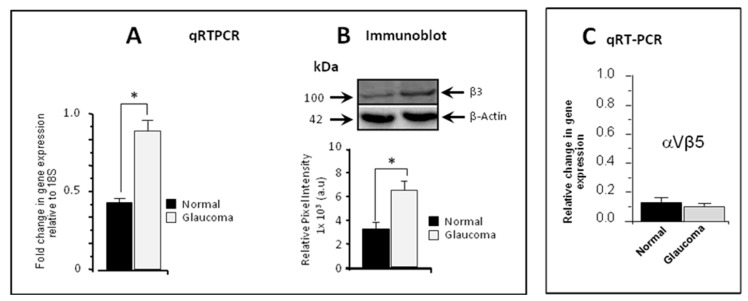
β3 integrin gene and protein expression in normal and glaucomatous LC cells. (**A**) Quantitative real-time RT-PCR illustrating β3 integrin gene transcription in normal vs. glaucoma LC cells, normalised to a housekeeping gene 18S. Note the significant enhancement of β3 integrin gene transcription levels in glaucoma (0.861 ± 0.055) when compared to normal LC cells (0.482 ± 0.038) (*p* < 0.05, n = 5). (**B**) Western blot analysis for β3 protein expression comparing normal to glaucomatous LC cells normalised to β-actin. There is a significant increase in protein expression levels in glaucoma (from 3.364 ± 0.534 to 6.477 ± 0.622) (10^3^ a.u) (*p* < 0.05, n = 5). (**C**) Quantitative RT-PCR illustrating β5 integrin gene expression. β5 integrin gene was expressed at very low levels and no significant differential expression was found when comparing normal to glaucoma LC cells. Significantly different values compared to controls are denoted by asterisks. Data were expressed as mean ± S.D. from five independent biological replicates; each experiment was performed in triplicate; * *p* < 0.05 vs. control.

**Figure 2 cells-13-01487-f002:**
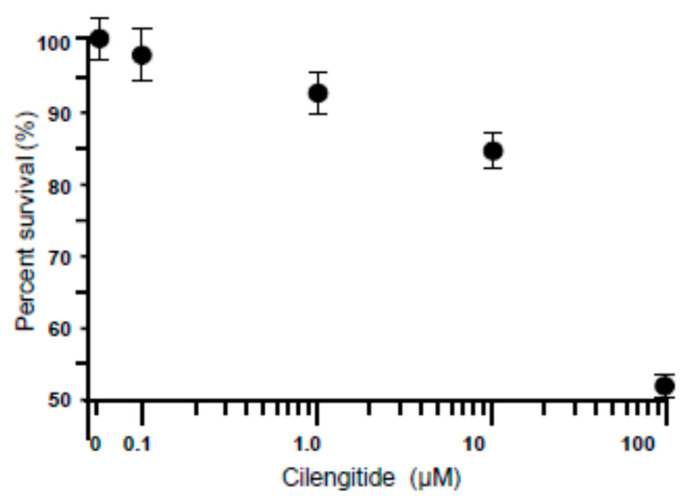
Dose-dependent effects of cilengitide on LC cell viability. Following 48 h of culture in a full DMEM-F12 medium, LC cells (10^4^/well) were serum-starved for 24 h, and then incubated for an additional 24 h in the presence (+) or absence (−) of increasing concentrations of cilengitide (0–100 µM) and then were analysed for cell viability. Cilengitide treatment had no effect on the number of viable cells at the concentrations of 0.1 µM to 10 µM. Note that 100 µM of cilengitide treatment resulted in a significant decrease of the number of viable cells. Therefore, a concentration of 10 µM was used for all subsequent experiments. Untreated cells were used as a control and the mean OD of the control was set to 100%. Data represent the mean ± SEM of five independent experiments (n = 5 different donors per group) performed in triplicate to ensure reproducibility of results.

**Figure 3 cells-13-01487-f003:**
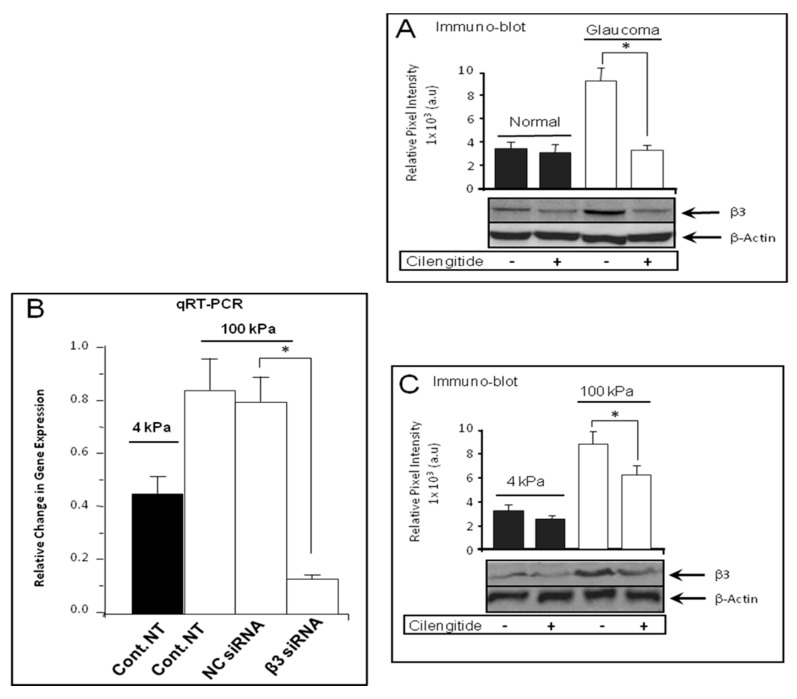
Cilengitide reduced β3 integrin expression in LC cells. LC cells from normal non-glaucomatous donors were cultured on soft (4 kPa) and stiff (100 kPa) matrix, in the presence (+) or absence (−) of cilengitide (10 µM). LC cells were grown in serum-free medium for 24 h and treated with cilengitide (10 µM) for an additional 24 h. (**A**) Western blot analysis showed that β3 integrin protein expression level was elevated in glaucoma LC cells, and cilengitide treatment significantly reduces this elevation (9.201 ± 1.246 (×10^3^ a.u) untreated vs. 3.424 ± 0.976 (×10^3^ a.u) treated) (*p* < 0.05, n = 5). Significantly different values are denoted by asterisks. Data were expressed as mean ± S.D. from five biological replicates; * *p* < 0.05 vs. control. (**B**) Quantitative real-time PCR analysis illustrating enhanced β3 integrin gene expression when normal LC cells were grown on a stiffer matrix (1.071 ± 0.135 (4 kPa) vs. 1.654 ± 0.174 (100 kPa)) (values expressed as fold change normalised to housekeeping ribosomal gene 18S), (*p* < 0.05, n = 5). Cilengitide (10 µM) treatment for 24 h resulted in a significant reduction of β3 integrin gene expression on stiff matrix (1.654 ± 0.174 untreated vs. 1.204 ± 0.097 treated) (*p* < 0.05, n = 5). (**C**) Western blot analysis illustrating a significant enhancement in protein expression level of β3 from soft to stiff matrix (3.638 ± 0.563 (×10^3^ a.u) (4 kPa) vs. 8.821 ± 1.246 (×10^3^ a.u) (100 kPa)) (*p* < 0.05, n = 5). Cilengitide treatment significantly reduced the stiffness-induced β3 protein expression levels from 8.821 ± 1.246 (×10^3^ a.u) (100 kPa, untreated) to 5.818 ± 0.984 (×10^3^ a.u) (100 kPa, treated).

**Figure 4 cells-13-01487-f004:**
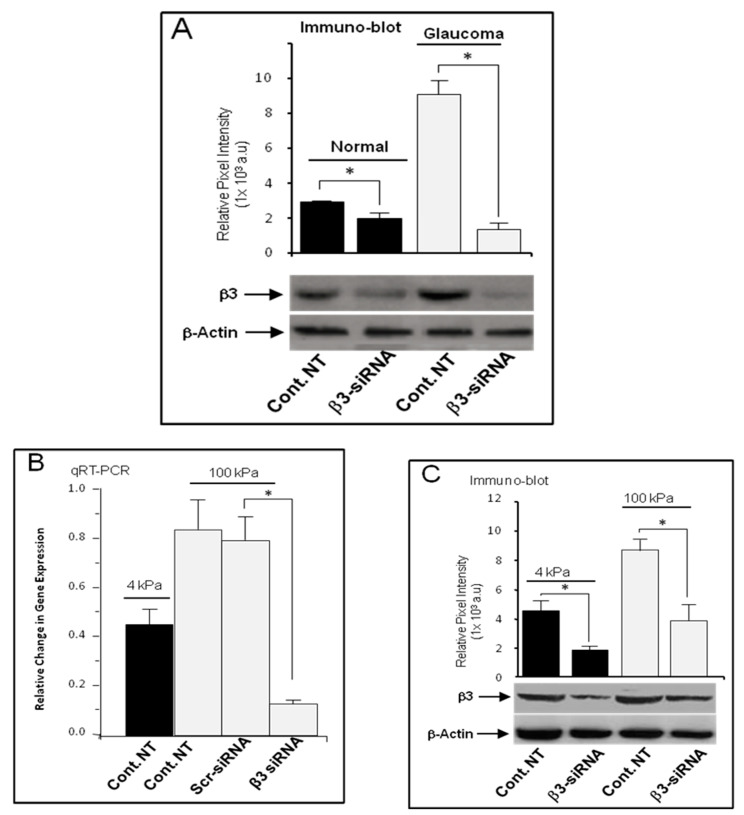
siRNA knockdown suppressed β3 integrin expression in LC cells. (**A**) LC cells from normal and glaucoma donors were grown in a full DMEM-F12 medium, in T25 flat-bottom plastic plates, until 50–60% confluence. Cells were first serum-starved for 24 h and then either transfected (treated group) or not (control group) with β3 integrin siRNA (50 nM) in serum- and antibiotics-free DMEM-F12 medium. Six hours later, the medium was replaced with full medium with antibiotics and serum, and protein extraction was performed at day 3 post-transfection. Western blot analysis indicates that β3 integrin protein expression level was elevated in glaucoma LC cells, and this elevation was knocked down by ~74% in LC cells transfected with 50 nM β3 siRNA compared to untransfected LC cells (*p* < 0.05, n = 5). In the second set of experiments, LC cells from non-glaucomatous donors were grown on soft (4 kPa) and stiff (100 kPa) matrices in similar experimental conditions as above. (**B**) qRT-PCR analysis illustrating mRNA levels of β3 integrin following 72 h post-transfection with siRNA against β3 integrin (treated group) compared with controls (untreated group). The stiffness-induced β3 integrin gene expression was inhibited by ~65% when compared to untreated LC cells (*p* < 0.05, n = 5), while the level of αVβ3 integrin gene expression level in LC cells transfected with scrambled non-coding siRNA remained unchanged. (**C**) Western blot analysis showed that the stiffness-induced β3 integrin protein expression level was reduced by ~45% when compared to the untreated control group. Data were expressed as mean ± S.D. from five biological replicates. Significantly different values are denoted by asterisks. (Cont.: control; NT: non-transfected; SRC: srcambled siRNA).

**Figure 5 cells-13-01487-f005:**
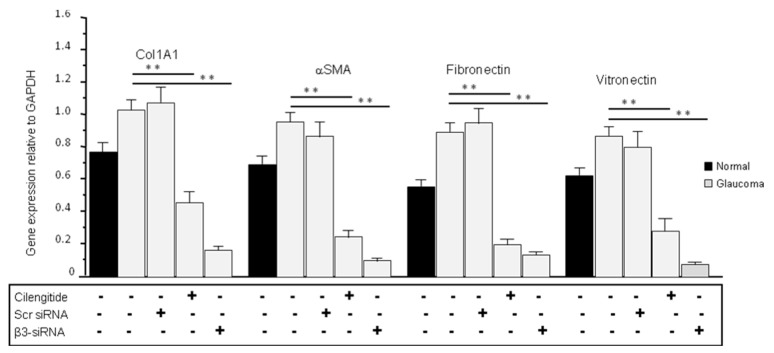
Cilengitide and β3 integrin siRNA treatments reduced the expression of pro-fibrotic ECM genes in glaucoma LC cells. Normal and glaucoma LC cells were grown on standard flat-bottom plastic plates and were either untreated (controls) or treated with cilengitide (10 µM) or transfected with an anti-β3 siRNA (50 nM) in the same experimental condition as in Figure 3 and Figure 4. Results showed that ECM gene transcription was significantly lower in untreated normal LC cells than in untreated glaucoma LC cells for all tested genes including Col1A1, αSMA, fibronectin, and vitronectin. Cilengitide treatment of glaucoma LC cells significantly reduced the ECM gene expression of Col1A1 (from 1.022 ± 0.073 to 0.453 ± 0.064), αSMA (from 0.948 ± 0.068 to 0.239 ± 0.038), fibronectin (from 0.892 ± 0.067 to 0.198 ± 0.039), and vitronectin (from 0.877 ± 0.065 to 0.283 ± 0.071) (*p* < 0.05, n = 5). Glaucoma LC cells were also transfected with siRNA targeting β3 integrin and the results, at day 3 post-transfection, show a clear significant inhibition of ECM gene transcription for Col1A1 (from 1.022 ± 0.073 to 0.168 ± 0.035), αSMA (from 0.948 ± 0.068 to 0.097 ± 0.022), fibronectin (from 0.892 ± 0.067 to 0.134 ± 0.0328), and vitronectin (from 0.877 ± 0.065 to 0.075 ± 0.021) (*p* < 0.05, n = 5). Significantly different values compared to stiff controls are denoted by asterisks. Data were expressed as mean ± S.D.; ** *p* < 0.02 vs. control.

**Figure 6 cells-13-01487-f006:**
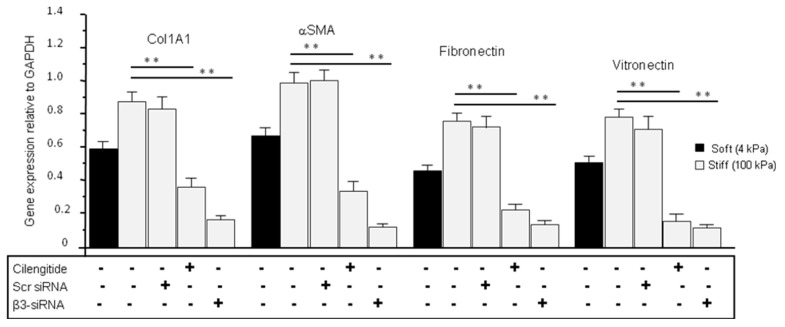
Cilengitide and β3 integrin siRNA reduced the stiffness-induced expression of pro-fibrotic ECM genes in normal LC cells. LC cells from normal non-glaucomatous patients were cultured on soft (4 kPa) and stiff (100 kPa) matrix, alone (-) or plus (+) cilengitide (10 µM) for 24 h in the same experimental conditions as in Figure 3A. Quantitative real-time RT-PCR illustrated that cilengitide treatment significantly reduced the stiffness-induced expression of pro-fibrotic markers Col1A1 (from 0.872 ± 0.097 stiff untreated to 0.364 ± 0.039 stiff treated), αSMA (from 0.982 ± 0.115 stiff untreated to 0.356 ± 0.044 stiff treated), fibronectin (from 0.763 ± 0.105 stiff untreated to 0.212 ± 0.036 stiff treated), and vitronectin (from 0.791 ± 0.108 stiff untreated to 0.175 ± 0.053 stiff treated) (*p* < 0.05, n = 5). In addition, the effect of siRNA β3 integrin silencing on ECM gene expression at day 3 post-transfection in NLC cells grown on stiff matrix was assessed. qRT-PCR results showed that stiffness induced a significant (*p* < 0.05) enhancement, while silencing β3 integrin significantly suppressed the stiffness-induced enhancement in gene transcription of Col1A1 (from 0.872 ± 0.097 stiff untreated to 0.174 ± 0.019 stiff treated), αSMA (from 0.982 ± 0.115 stiff untreated to 0.117 ± 0.015 stiff treated), fibronectin (from 0.763 ± 0.105 stiff untreated to 0.191 ± 0.017 stiff treated), and vitronectin (from 0.791 ± 0.108 stiff untreated to 0.183 ± 0.013 stiff treated) (*p* < 0.05, n = 5). Significantly different values compared to stiff controls are denoted by asterisks. Data were expressed as mean ± S.D.; ** *p* < 0.02 vs. control.

**Figure 7 cells-13-01487-f007:**
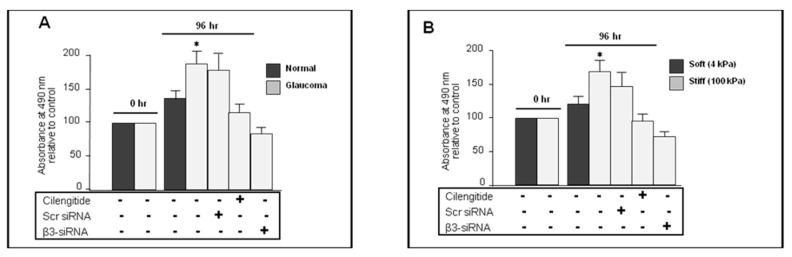
Cilengitide and β3 integrin siRNA silencing inhibited LC cell proliferation. LC cells seeded at a density of 10^4^ cells/well were either cultured on flat-bottom standard plastic 96-well plates or on soft (4 kPa) and stiff (100 kPa) matrix. Treatments with cilengitide and siRNA were performed in the same experimental conditions as in Figure 3 and Figure 4. (**A**) Normal and glaucoma LC cells were grown on 96-well flat-bottom plastic plates. MTS assay showed a significant increase in cell proliferation rate in untreated glaucoma LC cells (187.31 ± 21.14% vs. 135.46 ± 14.70 in untreated normal LC cells), while cilengitide treatment of glaucoma cells significantly reduced the cell proliferation rate to 116.182 ± 12.42% (*p* < 0.05, n = 5). Similarly, siRNA silencing β3 integrin significantly suppressed LC cell proliferation rate in glaucoma from 187.31 ± 21.14% in untreated glaucoma compared to 83.42 ± 8.44% in treated glaucoma LC cells (*p* < 0.05, n = 5). (**B**) Normal LC cells grown on soft (4 kPa) and stiff (100 kPa) matrix, as measured with MTS assays, revealed a significant increase in proliferation rate at 100 kPa (untreated stiff 171.61 ± 16.59% vs. untreated soft 122.82 ± 10.73%), and cilengitide treatment significantly reduced cell proliferation rate (stiff untreated 171.61 ± 16.59% vs. stiff treated 97.32 ± 8.70%) (*p* < 0.05, n = 5). Stiffness-induced proliferation was also examined using siRNA β3 integrin silencing. siRNA β3 integrin silencing significantly inhibited the stiffness-induced proliferation rate (stiff untreated 171.61 ± 16.59% vs. stiff treated 72.63 ± 7.94%) (*p* < 0.05, n = 5). Significantly different values compared to untreated are denoted by asterisks. Data represent the mean ± SD of five independent experiments performed in triplicate. * *p* < 0.05 vs. control.

**Table 1 cells-13-01487-t001:** Posterior sclera specimens from human donor eyes with diagnosed glaucoma and without glaucoma (normal).

Donor ID	Age	Gender	Disease State	Eye Bank
41-02	88	F	Non-glaucoma	Central Florida (Tampa)
82-02	88	F	Non-glaucoma	Central Florida (Tampa)
439-02	79	M	Non-glaucoma	Central Florida (Tampa)
444-02	87	M	Non-glaucoma	Central Florida (Tampa)
553-02	87	F	Non-glaucoma	Central Florida (Tampa)
58-02	84	F	Glaucoma	Central Florida (Tampa)
350-02	84	F	Glaucoma	Central Florida (Tampa)
412-02	85	M	Glaucoma	Central Florida (Tampa)
600-02	86	M	Glaucoma	Central Florida (Tampa)
652-02	79	M	Glaucoma	Central Florida (Tampa)

**Table 2 cells-13-01487-t002:** Pro-fibrotic and αv/β3/β5 integrin and ECM primers used for real time qRT-PCR.

Human Genes	Forward	Reverse
Integrin αv	AATCTTATTGAGGATATCAC	AAAACAAGTAGCAACAAT
Integrin β5	GGAGCCAGAGTGTGGAAACA	GAAACTTTGCAAACTCCCTC
Integrin β3	GTCACCTGAAGGAGAATCTGC	TTCTTCGAATCATCTGGCC
αSMA	CCGACCGAATGCAGAAGGA	ACAGAGTATTTGCGCTCCGAA
Col1A1	ACGAAGACATCCCACCAATC	ATGGTACCTGAGGCCGTTC
Fibronectin	ACAACACCGAGGTGACTGAGAC	GGACACAACGATGCTTCCTGAG
Vitronectin	ATGGGTTGCTCTGGCTGAC	CTGCTGGGGGCTGAGGTCT
18S	CTGGGACGACATGGAGAAAA	AAGGAAGGCTGGAAGAGTGC

## Data Availability

The original data presented in the study are openly available.

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
