# Peer review of "The Role of αvβ3 Integrin in Lamina Cribrosa Cell Mechanotransduction in Glaucoma"

_cells, 2024, doi:10.3390/cells13171487_

Round 1

Reviewer 1 Report

Comments and Suggestions for Authors

The present study evaluated whether there is a role of αVβ3 integrin in glaucoma. The authors found over-expression of the αVβ3 integrin in the LC cells of post-mortem glaucomatous eyes, and the administration of cilengitide αVβ3 integrin siRNA treatment significantly reduced this over-expression.

The study concluded that treatments with cilengitide or siRNA mediated knockdown of the αVβ3 integrin could be a potential filed of interest in the novel therapy for glaucoma.

The study is novel, well-conducted, and potentially useful for both readers and for further studies. Some points to address are the followings.

The authors considered five normal and five affected, age matched, glaucomatous post-mortem eyes. 

Please, discuss whether there are age-related changed in the expression of integrins on surface of LC cells. This is a probably important topic since, while glaucoma is more frequent in older subjects, it expresses more aggressively, in many cases, in younger cases. 

Thus, studies with greater sample permitting to perform an age-based analysis are mandatory.

As the authors know, the corneal hysteresis is a surrogate biomarker of the lamina cribrosa stiffness. To date, with currently available diagnostic platforms, we can precisely measure the corneal hysteresis, but we are unable to measure the LC hysteresis. 

Given that these molecular aspects are probably the molecular background of what we clinically measure, that is the corneal hysteresis, to further support the results of this study, it can be useful to discuss whether there are integrins modifications also within the cornea in glaucoma.

Cilengitide seems a promising arm to counteract the LC stiffness. It has been used in patients with SNC neoplasia. Does literature present studies that evaluated corneal hysteresis, or potentially useful clinical effects, in patients concomitantly affected with glaucoma? 

Did literature report some evidence of the involvement of the αVβ3 integrin in other neuro-degenerative diseases such as glaucoma? Was cilengitide and siRNA mediated knockdown of the αVβ3 integrin used to treat other neuro-degenerative diseases?

Discussion section should be improved including a limitation section.

Are there potential side effects of treatments using αVβ3 integrin inhibitors?

Comments on the Quality of English Language

The English presentation needs some minor improvements

Reviewer 2 Report

Comments and Suggestions for Authors

The author cultured primary lamina cribrosa (LC) cells and investigated the role of integrins in normal or glaucoma LC cells. This study is innovative and presented clearly.

The only concern is figure labeling. First, please replace figures with higher resolution format (e.g.: TIFF) to show clearer labeling. Second, Figure 1A Immunoblot did not label the normal or glaucoma LC. In Figure 3B, it is difficult to tell the difference between 100kPa and 100kPa + Cilengitide. Please change one of them to a white bar. 
